Heeding the psychological concerns of young cancer survivors: a single-arm feasibility trial of CBT and a cognitive behavioral conceptualization of distress

Hagström Josefin 1
Ander Malin 1
Cernvall Martin 1
Ljótsson Brjánn 2
Wiman Henrik W. 1
von Essen Louise 1
Woodford Joanne joanne.woodford@kbh.uu.se 1
1 Clinical Psychology in Healthcare, Department of Women’s and Children’s Health, Uppsala Universitet , Uppsala , Sweden
2 Division of Psychology, Department of Clinical Neuroscience, Karolinska Institute , Stockholm , Sweden
Zeppel Melanie
Electronic publication date: 2020 Mar 19
Publication date: 2020
Volume: 8
Electronic Location ID: e8714
Received 2019 May 16; Accepted 2020 Feb 10
Copyright: ©2020 Hagström et al.
Copyright year: 2020
Copyright holder: Hagström et al.
License: This is an open access article distributed under the terms of the Creative Commons Attribution License, which permits unrestricted use, distribution, reproduction and adaptation in any medium and for any purpose provided that it is properly attributed. For attribution, the original author(s), title, publication source (PeerJ) and either DOI or URL of the article must be cited.
License URL: https://creativecommons.org/licenses/by/4.0/

Keywords: AYA survivor, Feasibility, Cancer, CBT, Behavioral conceptualization, Psychological concerns, Internet-based intervention

Funding: Swedish Childhood Cancer Foundation PR2013-0039 PR2016-0004 PR2017-0005 The Swedish Cancer Society 13 0457 Swedish Research Council to the strategic research program U-CARE This work was supported by the Swedish Childhood Cancer Foundation, grant numbers PR2013-0039; PR2016-0004 and PR2017-0005, the Swedish Cancer Society, grant number 13 0457, and funding via the Swedish Research Council to the strategic research program U-CARE. The funders had no role in study design, data collection and analysis, decision to publish, or preparation of the manuscript.

==============================
Background

A subgroup of adolescent and young adult (AYA) survivors of cancer during adolescence report high levels of psychological distress. To date, evidence-based psychological interventions tailored to the cancer-related concerns experienced by this population are lacking. The present study aimed to (1) examine the feasibility and preliminary efficacy of an individualized cognitive behavioral therapy (CBT) intervention for AYA survivors of cancer during adolescence; and (2) identify and conceptualize cancer-related concerns as well as maintaining factors using cognitive-behavioral theory.

Methods

A single-arm trial, whereby AYA survivors of cancer during adolescence (aged 17–25 years) were provided individualized face-to-face CBT at a maximum of 15 sessions. Clinical outcomes were assessed at baseline, post-intervention, and three-month follow-up. Intervention uptake, retention, intervention delivery, and reliable change index scores were examined. An embedded qualitative study consisted of two unstructured interviews with each participant pre-intervention. Along with individual behavioral case formulations developed to guide the intervention, interview data was analyzed to identify and conceptualize cancer-related concerns and potential maintaining factors.

Results

Ten out of 213 potential participants invited into the study were included, resulting in an overall participation rate of 4.7%. Nine participants completed the intervention, with respectively seven and eight participants completing the post-intervention and three month follow-up assessment. The majority of reported cancer-related concerns and maintaining factors were conceptualized into four themes: social avoidance, fear of emotions and bodily symptoms, imbalance in activity, and worry and rumination.

Conclusions

Given significant recruitment difficulties, further research is required to examine barriers to help-seeking in the AYA cancer survivor population. However, the conceptualization of cancer-related concerns and maintaining factors experienced by the population may represent an important first step in the development of psychological support tailored toward AYA cancer survivors’ unique needs.

Introduction

Adolescent and young adult (AYA) survivors of cancer diagnosed during adolescence report numerous stressors relating to cancer diagnosis, treatment, and survivorship (Abrams, Hazen & Penson, 2007; Lehmann et al., 2014). The adolescent period includes challenges in terms of development as well as transition into adulthood (Jaworska & MacQueen, 2015), and the onset of several common mental health difficulties (Kessler et al., 2012). As such, AYAs diagnosed with cancer during adolescence appear at an increased risk of problems related to their mental health. Indeed, studies have demonstrated that AYA survivors of cancer during adolescence report elevated levels of psychological distress in comparison to matched cancer-free controls (Seitz et al., 2010), those diagnosed during childhood (Kazak et al., 2010) and adulthood respectively (Lang et al., 2018). Further, research indicates that receiving a cancer diagnosis prior to age 25 leads to a higher risk of suicide (Gunnes et al., 2017) and being prescribed anxiolytics and hypnotics (Johannsdottir et al., 2018). Although some evidence has suggested lower levels of psychological distress compared to population norms (Larsson, Mattsson & Von Essen, 2010), research has consistently identified a subgroup of AYA survivors of cancer during adolescence who report an elevated level of long-term psychological distress (De Laage et al., 2016; Husson et al., 2017).

Despite a significant subgroup of AYA survivors of cancer during adolescence reporting an elevated level of psychological distress, the group reports unmet needs of psychological support (Bibby et al., 2017; Kaul et al., 2017). As a result, the development of psychological interventions adapted for the unique concerns of this group has been posited (Ander et al., 2016; Seitz, Besier & Goldbeck, 2009; Seitz et al., 2010). One potential solution may be the provision of psychological support based on cognitive-behavioral therapy (CBT), an evidence-based approach that has been demonstrated to be effective for a range of mental health difficulties (Kaczkurkin & Foa, 2015; McMain et al., 2015). Indeed, CBT-based interventions have shown promise in reducing psychological distress in AYAs diagnosed with cancer during childhood (Fisher et al., 2015; Seitz et al., 2014; Van der Gucht et al., 2017). Whilst some recognition has been given the specific and unique challenges associated with being diagnosed with cancer during adolescence (Abrams, Hazen & Penson, 2007; Seitz, Besier & Goldbeck, 2009), so far only a few CBT based psychological interventions have been developed with adaptation and tailoring to this population (Campo et al., 2017; Sansom-Daly et al., 2018).

In order to overcome this psychological treatment gap, a thorough theoretical understanding is needed concerning symptoms of psychological distress and mechanisms involved in the maintenance of distress (Craig et al., 2008). Two models have been proposed to guide the development of psychological interventions for adolescents diagnosed with cancer: (1) the Adolescence Resilience Model (Haase, 2004) and (2) the Pediatric Medical Traumatic Stress Model (Kazak et al., 2006). However, neither model recognizes the full range of areas of cancer-related distress experienced by AYA survivors of cancer during adolescence, such as feelings of worthlessness and feeling left behind (Ander et al., 2018). Nor does either model specify the concerns or mechanisms hypothesized to be involved in the maintenance of distress. Further, psychological interventions based on these models have shown limited effect on the reduction of cancer-related distress (Kazak et al., 2004; Robb et al., 2014). Research to identify mechanisms involved in the maintenance of distress has started (Sansom-Daly & Wakefield, 2013); however, there is a need to further extend the knowledge of the different manifestations of psychological distress experienced by the population, alongside an examination of cancer-related concerns and maintaining factors for distress. Such knowledge can be used to guide development of psychological interventions tailored for AYA survivors of cancer during adolescence (Seitz et al., 2010).

Seeking to inform the development of a CBT-based intervention tailored for AYA survivors of cancer during adolescence, the present study had a twofold primary aim: (1) to examine the feasibility (e.g., recruitment, attrition, data collection, intervention procedures) and preliminary efficacy of an individualized CBT intervention for the population; and (2) to develop a cognitive-behavioral theory-based conceptualization of cancer-related psychological concerns and maintaining factors experienced by the population.

Materials & Methods

Study design

A single-arm trial design with a pre-post and three month follow-up design, and embedded qualitative study, was adopted to examine the feasibility and preliminary efficacy of individualized CBT and develop a conceptualization of cancer-related psychological distress. The study was approved by the Uppsala Regional Ethical Review Board (reference number: 2014/443).

Setting

Delivery of individualized CBT and data collection was undertaken at two locations: the pediatric oncology unit at the Children’s University Hospital in Uppsala and a private psychology practice in Stockholm.

Participants

Participants were eligible if: (1) aged between 15–25 years; (2) diagnosed with cancer during adolescence (13–19 years); (3) completed cancer treatment at the pediatric oncology center in Uppsala or Stockholm (according to pediatric oncology centers for persons 15–17 years, or self-report for persons ≥18 years); (4) able to participate in individual CBT once a week for up to 15 weeks in Uppsala or Stockholm; and (5) a self-reported need for psychological support. Potential participants were excluded if: receiving a psychological intervention; and/or a self-reported severe and/or enduring mental health problem (e.g., bipolar disorder, psychosis) and/or acute suicidality requiring more specialized treatment than could be offered within the present study.

Recruitment and study procedures

Recruitment took place from February through October 2015. First, potential participants diagnosed with cancer during adolescence, meeting the inclusion criteria, were identified via the Swedish Childhood Cancer Registry. Second, their telephone numbers were retrieved either via internet search engines (18–25 years) or pediatric oncology centers (15–17 years). All potential participants, whereby a telephone number could be identified, were telephoned by one of the research team members in two waves: first potential participants treated in Uppsala; and thereafter potential participants treated in Stockholm. In cases whereby the individual could not be invited via telephone a study invitation letter was sent via post, including a link to a study website where one could register contact details if interested in participating. Study invitation letters informed potential participants that the study was to examine CBT for AYA survivors of cancer during adolescence who currently experienced psychological difficulties related to the cancer experience. Potential participants were informed that the intervention would last 10–15 weeks and that therapeutic work would focus on helping them with difficult thoughts, feelings and behaviors.

All potential participants expressing interest to participate were invited to a face-to-face screening assessment with a study therapist (two licensed psychologists and one psychologist in clinical training). The Montgomery Åsberg Depression Rating Scale-Self Assessment (MADRS-S) (Svanborg & Åsberg, 1994) and Mini International Neuropsychiatric Interview (M.I.N.I.) version 5.0 (Sheehan et al., 1998) were administered to screen for depression and anxiety disorders (including PTSD) and to identify individuals with a severe or enduring mental health problem (e.g., bipolar disorder, psychosis) and/or acute suicidality. Individuals with such difficulties were excluded and guided to appropriate support. For one participant who was under the age of 18 at the study start, parents were provided with written information and informed about the study via telephone. In addition, an assessment was conducted of the participant’s understanding of the study to determine if there was a need for parental consent.

Written information about study participation was provided to potential participants prior to the screening assessment. If found eligible, study participation was offered and written informed consent obtained. Subsequently, baseline assessments, and two unstructured interviews were conducted. The screening, baseline assessments, and interviews took place over one-to-three sessions. After completion of the baseline assessments, participants were assigned to one of three study therapists. Two of the three study therapists were co-authors (MA and MC). In eight cases, the same study therapist who conducted the screening and baseline assessment also conducted subsequent interviews and delivered the intervention. Except for one case, post- and follow-up assessments were conducted by another study therapist than the one providing the intervention.

Sample size

As a primary aim was to examine feasibility (e.g., recruitment, data collection, intervention delivery), a formal sample size calculation was not conducted (Billingham, Whitehead & Julious, 2013). Study recruitment was stopped when all individuals identified via permission by the ethical committee had been screened for eligibility. Few psychological intervention studies have been conducted among AYA survivors of cancer during adolescence (Barnett et al., 2016) and estimations of recruitment rates were difficult to determine a priori. However, as informed by large mental health intervention trials in the general population, a recruitment rate of 12% may have been anticipated (Richards et al., 2013; Richards et al., 2016). At the time of the trial, 213 potentially eligible AYA cancer survivors were registered in the Swedish Childhood Cancer Registry in the Stockholm and Uppsala areas. Taking the estimate of a 12% recruitment rate, a sample size of 26 had been anticipated.

Data collection

Sociodemographics: Data for the following variables was collected via the Swedish Childhood Cancer Registry: (1) current age; (2) age at diagnosis; (3) gender; and (4) cancer diagnosis. Self-reported information at baseline assessment included: (1) cancer recurrence; (2) time since end of cancer treatment; (3) type of cancer treatment; (4) number and type of late effects; (5) previous receipt of psychological intervention; (6) accommodation status; (7) relationship status; and (8) employment status.

Feasibility outcomes: Feasibility outcomes related to recruitment, attrition, data collection, and intervention procedures were collected.

Clinical outcomes: A number of self-report clinical outcome measurements (Swedish translations) were administered at baseline, post-intervention, and three-month follow-up. The MADRS-S and the M.I.N.I. version 5.0 were re-administered at post-intervention and three-month follow-up.

Symptoms of anxiety were assessed using the 21-item Beck Anxiety Inventory (BAI) (Beck et al., 1988). Posttraumatic stress symptoms were assessed using the 17-item PTSD Checklist—Civilian Version (PCL-C) (Weathers et al., 1993) as defined in the B (re-experiencing), C (avoidance), and D (hyperactivity) criteria in DSM-IV (American Psychiatric Association, 2000), with items modified to focus on the participant’s experience of cancer. The 16-item Penn State Worry Questionnaire (PSWQ) (Meyer et al., 1990) was used to assess worry, and the 10-item Body Image Scale (BIS) (Hopwood et al., 2001) to measure body dissatisfaction. Health anxiety was assessed using the 18-item Short Health Anxiety Inventory (SHAI) (Salkovskis et al., 2002) and the Patient Health Questionnaire-15 (PHQ-15) (Kroenke, Spitzer & Williams, 2002) was adopted to measure somatic symptoms. The 10-item Fatigue Assessment Scale (FAS) (Michielsen, De Vries & Van Heck, 2003) was used to assess fatigue. Rumination was assessed using the 22-item Rumination Scale of the Response Style Questionnaire (RSQ) (Nolen-Hoeksema, 1991) and experiential avoidance was examined using the 10-item version of the Acceptance and Action Questionnaire (AAQ-II) (Bond et al., 2011). Functional impairment in work, social, and family life was measured using the three-item Sheehan Disability Scale (SDS) (Sheehan, 1983).

Unstructured interviews: Prior to commencing the intervention, two unstructured interviews (lasting 33 to 104 min) were conducted with each participant over a two-week period by the therapist the participant had been assigned to. All interviews began with the question: “Please tell me how you think and feel about having had cancer?.” Subsequently, probes such as “Tell me more about…”, “Tell me how you felt when…”, and “Please tell me what you mean by…” were used to encourage the participant to provide more in depth responses and facilitate clarification (Ander et al., 2018). Before the second interview, the interviewer listened to a recording of the first interview to identify areas for further exploration. To improve interview quality, a senior researcher not otherwise associated with the study, with extensive experience in qualitative interviewing, reviewed one interview per therapist.

Intervention

The intervention was delivered by two licensed psychologists and one psychologist in clinical training across the two sites. Given the explorative nature of the study, cognitive-behavioral case formulations were developed by study therapists, including the identification of current and previous problems and concerns, prioritization of concerns and specification of topography (i.e., descriptions of the specific behavioral features). The subsequent intervention was tailored to each participant’s individual problems and needs according to these case formulations (Persons, 2008). Further, functional analyses were developed using learning theory, to make explanatory inferences partly on the association between precipitating events and resulting problems, and partly on factors hypothesized to have maintained individuals’ problems (Dougher, 2000; Sturmey, 2008).

An initial working cognitive-behavioral case formulation was developed over intervention sessions one-to-three, and subsequently tested and refined during the course of the intervention. Study therapists used relevant cognitive-behavioral models and standard CBT techniques dependent on the main problems raised by participants. CBT techniques were discussed among study therapists during continuous supervision however individual study therapists made final decisions concerning the selection of CBT techniques. An overview of delivered techniques can be seen in Table 1.

Table 1 Number of participants receiving the respective CBT-based treatment techniques in treatment.

Treatment technique	n	
Functional analysis	10	
Goal setting	10	
Psychoeducation including experiential exercises	10	
Setting a maintenance plan	9	
Exploring values	8	
Self-monitoring	8	
Mindfulness-based exercises	7	
Value-guided behavior change/exposure relating to social avoidance and fear of failure	6	
Worry time, worry exposure, problem-solving	5	
Behavioral activation	3	
General affect exposure	3	
Stress management techniques	3	
Emotion regulation/distress tolerance training	2	
Exposure (interoceptive exposure and exposure & response prevention)	2	
Interpersonal effectiveness skills training	2	
Relaxation exercises	1	

Number of participants receiving the respective CBT techniques

Each participant was offered a maximum of 15 sessions of individual CBT (each session lasting approximately 45 min). Cognitive-behavioral case formulations was discussed with the other study therapists, in supervision by a licensed psychologist with expertise in delivering CBT to young people experiencing mental health problems and clinical behavior analysis, and in external supervision of a licensed psychologist with expertise in clinical behavior analysis. Both supervisors were external to the study team. Formulations were also discussed during internal supervision to further receive opinions and increase the validity of the formulations

Data analysis

An adapted version of the Consolidated Standards of Reporting Trials (CONSORT) diagram for parallel randomized trials (Moher et al., 2010) was used to illustrate participant flow. Descriptive statistics (means and SDs) were used to report clinical outcomes at baseline, post-treatment, and three-month follow-up assessments. Descriptive statistics were used to report feasibility outcomes pertaining to recruitment, attrition, data collection, and treatment delivery. As a preliminary investigation of efficacy, the proportion of participants reporting a reliable change in scores on the MADRS-S, BAI, and/or PCL-C was calculated according to the Jacobson-Truax Index (Jacobson & Truax, 1991), with test-retest reliability of the measures from previous studies obtained (Fydrich, Dowdall & Chambless, 1992; Ruggiero et al., 2003; Svanborg & Åsberg, 1994) to inform the calculation. Missing items were imputed using the mean item score of available items on the outcome measurement in cases whereby a maximum of two, or 10%, of items in a measurement, were missing. List-wise deletion was used for the calculation of RCI for the cases whereby baseline-, post-, or three month follow-up assessment was missing. As such, participants were excluded from analysis if their post-intervention and/or three-month follow-up assessment was missing.

No formal method of analyzing the qualitative interview data was adopted. However, the method was informed by steps utilized in qualitative framework analysis (Ritchie & Spencer, 1994) and theoretical thematic analysis (Braun & Clarke, 2006). A framework was developed a priori for extracting and summarizing the data for the purpose of categorizing cancer-related and non-cancer-related concerns and key characteristics informed by CBT (e.g., current and previous concerns, topography, maintaining factors). Moreover, informal discussions and reflections regarding the conceptualization and emerging themes were ongoing among therapists throughout the study.

First, with the aim of identifying current and previous cancer-related concerns alongside, topography and maintaining factors, interview transcripts, clinical notes, and behavioral case formulations were read by the second author (MA) to gain a sense of the dataset. Behavioral case formulations were thereafter read by MA to identify cancer-related concerns which were refined into themes and defining key characteristics informed by CBT. Thereafter, interview transcripts were re-read by MA to extract cancer-related concerns considered relevant for each theme. The third author (MC) reviewed the themes of cancer-related concerns and experiences with respect to coherence, congruence, and distinctiveness (Braun & Clarke, 2006) and themes were subsequently elaborated and refined. CBT case formulations and clinical notes were further examined by MA, to identify and summarize the specific CBT techniques applied during the intervention (see Table 1). Themes were finally reviewed, alongside the original data by MC to further increase trustworthiness.

Results

Demographic and clinical outcomes

Participant characteristics are provided in Table 2. According to the cancer registry, the majority suffered from leukemia (50%). Most reported living with their parents (60%); no history of relapse (90%); having experienced late effects (80%); and previous receipt of psychological support at some point in life (69%). One participant self-reported being diagnosed with Myelodysplastic syndrome (MDS) which contrasted with registry information.

Table 2 Participant characteristics (n = 10).

	Mean	SD	Range	
Age at study-start (years)	21	2.9	17–25	
Age at cancer diagnosis (years)	15.9	1.4	13–17	
Time since end of cancer treatment (years) (n = 9)	3.2	1.7	1–6	
	n			
Female/Male	4/6			
Cancer diagnosis from self-report				
Leukemia	4			
CNS-tumor	2			
Lymphoma	2			
Soft tissue sarcoma	1			
Other malignancy	1			
Cancer diagnosis from registry				
Leukemia	5			
CNS-tumor	2			
Lymphoma	2			
Soft tissue sarcoma	1			
Other malignancy	0			
History of relapse yes/no	1/9			
Living situation				
With parents	6			
Not with parents	4			
Relationship status				
Single	5			
In a relationship	5			
Employment status				
Student	6			
Employed full-time	1			
Employed part-time	2			
Unemployed	1			
Sick leave	0			
Self-reported late effects yesa /no	8/2			
Previous psychological treatment yes/no	6/4			
Notes.

N/n number of participants

SD Standard Deviation

a Self-reported late effects included anemia, balance problems, fatigue/tiredness, headaches, high blood pressure, kidney stone, loss of appetite, memory problems, nausea, nerve damage, nerve pain, numbness, speech difficulties, and thrombus.

Means, standard deviations for clinical outcomes at baseline, post-treatment, and three-month follow-up assessment are summarized in Table 3.

The proportions of participants meeting criteria for reliable change on the MADRS-S, BAI, and PCL-C were calculated for seven participants providing complete outcome data at post-intervention and three-month follow-up and are presented in Table 4. At post-intervention, 3/7 (43%) participants met criteria for reliable change on at least one outcome, with 4/7 (57%) meeting criteria at three-month follow-up. Except for one participant’s score on the PCL-C, all participants reported reduced scores at follow-up compared to baseline on the self-report measures.

Feasibility outcomes

A summary of feasibility outcomes is shown in Table 5, with more detailed data concerning the main feasibility objectives reported below.

Table 3 Means, standard deviations, and ranges for the clinical outcome measurements at baseline, post-treatment, and three month follow-up assessment (n = 10).

	Baseline (n = 9)	Post (n = 7)	Follow-up (n = 8)	
	Mean	SD (range)	Mean	SD (range)	Mean	SD (range)	
MADRS-S	17.0	6.2 (8–27)	10.9a	7.6 (3–26)	9.8	7.1 (3–25)	
BAI	14.6	16.5 (1–56)	7.9a	6.5 (2–21)	6.1	8.0 (0–25)	
PCL-C	37.0	14.2 (20–70)	27.1a	8.1 (20–45)	27.4	11.7 (18–55)	
PSWQ	48.9	11.5 (32–63)	40.9a	5.7 (32–49)	38.8	9.3 (22.4–52)	
AAQ-II	36.3	10.6 (23–55)	32.4a	10.3 (19–47)	28.6	12.1 (17–50)	
BIS	13.2	7.9 (5–29)	8.0	4.1 (4–14)	8.8	8.1 (0–25)	
FAS	26.8	5.6 (20–36)	22.3	5.5 (17–31)	21.5	5.6 (12–29)	
PHQ-15	9.5	6.7 (2–23)	6.1	3.6 (1–12)	6.8	4.1 (2–14)	
R-RSQ	53.0	14.1 (30–72)	47.8a	17.5 (25–74)	44.0	15.2 (24–70)	
SDS	12.9	5.9 (4–21)	7.5a	6.1 (0–15)	5.8	5.5 (0–15)	
SHAI Main	14.8	4.6 (8–20)	11.7	4.6 (4–17)	8.0	3.2 (1–12)	
SHAI Negative consequences	3.4	2.7 (0–7)	2.3	2.2 (1–6)	2.9	1.9 (0–5)	
SHAI-Avoidance	14.0	14.3 (0–42)	8.1	9.6 (0–22)	7.6	8.2 (0–19)	
SHAI Reassurance	15.1	6.6 (6–24)	11.9	9.3 (1–23)	7.3	6.1 (0–17)	
Notes.

a n =8.

n number of participants

SD Standard Deviation

MADRS-S Montgomery Åsberg Depression Rating Scale –Self Assessment

BAI Beck Anxiety Inventory

PCL-C The PTSD Checklist-Civilian Version

PSWQ Penn State Worry Questionnaire

AAQ-II Acceptance and Action Questionnaire-II

BIS Body Image Scale

FAS Fatigue Assessment Scale

PHQ-15 Public Health Questionnaire-15

R-RSQ Rumination Scale of the Response Style Questionnaire

SDS Sheehan Disability Scale

SHAI The Short Health Anxiety Inventory

Recruitment: A CONSORT diagram, adapted for a single-armed trial, illustrating the flow of participants through the study, including reasons for non-participation, is shown in Fig. 1.

Out of 286 initially identified individuals via the Swedish Childhood Cancer Registry, 213 potential participants (213/286; 75%) were to be contacted, with the remaining 73 individuals excluded (see Fig. 1 for reasons). Of those, 209 were contacted and received study information via telephone (111/209; 53%), or the post (98/209; 47%). Nine of those invited via telephone (9/111; 8%) and four of those invited via the post (4/98; 4%) expressed interest in study participation and were assessed for eligibility (in total: 13/213, 6%). Of those assessed for eligibility, three were excluded (3/13; 23%) while ten (10/13; 77%) were eligible and allocated to the intervention, resulting in an overall participation rate of 5% (10/213).

Attrition: Out of the ten participants allocated to the intervention, one participant discontinued participation before receipt of the intervention, resulting in a total of nine participants (9/10; 90%) receiving the intervention. Completion rates of assessments were as follows: baseline assessment, n = 9; post-intervention assessment, n = 7; and three month follow-up assessment, n = 8.

Table 4 Number of individuals with complete data reaching reliable change from baseline to post-assessment and from baseline to follow-up assessment (n = 7).

	Baseline/Post-assessment	Baseline/Follow up assessment	
MADRS-S	2	3	
PCL-C	2	2	
BAI	1	1	
BAI, PCL-C and/or MADRS-S	3	4	
Notes.

n number of participants

MADRS-S Montgomery Åsberg Depression Rating Scale–Self Assessment

PCL-C The PTSD Checklist-Civilian Version

BAI Beck Anxiety Inventory

Reliable change scores were 7.7 for MADRS-S, 13.6 for PCL-C, and 22.9 for BAI.

Data collection: The last post-intervention assessment was conducted in April 2016 and the last three month follow-up assessment in June 2016. The mean time between post-intervention and follow-up assessment was 12 weeks (ranging from 10 to 14 weeks). In total, nine items were missing from the self-report questionnaires (one in PHQ-15, two in SHAI-Reassurance subscale at baseline; two in BAI, one in BIS and one in PHQ-15 at post-intervention; one in AAQ-II and one in PHQ-15 at follow-up). Given the low rates of missing items, missing items were imputed using the mean item score of available items on the respective questionnaire for that participant at that specific assessment.

Intervention delivery: Number of intervention sessions ranged from 7 to 15 (mean = 13 sessions, median = 14 sessions) with a mean intervention length of 25 weeks (range = 18–36 weeks), including summer and winter holidays and cancelled intervention sessions. In two cases, the intervention was terminated early despite participants still reporting distress. In one case, the study therapist was unable to complete the intervention and the participant did not want a new study therapist. In the second case, the intervention was terminated due to poor session attendance combined with an identified need for more specialized intervention. The number of cancelled intervention sessions per participant ranged from 0 to 11 (mean = 3.4 sessions, median = 2 sessions).

Cognitive behavioral conceptualization of cancer-related concerns

A conceptualization was developed, encompassing four themes of cancer-related concerns: social avoidance (n = 7), fear of emotions and bodily symptoms (n = 7), imbalance in activity (n = 8), and worry and rumination (n = 8) (the latter was to some extent included within the other themes, yet is presented separately due to being a salient target in the intervention delivered). Each theme is presented in Fig. 2 along with defining key characteristics and examples of cancer-related concerns and maintaining factors.

In addition, a de-identified worked example of an individual’s case formulation can be seen in Fig. 3, demonstrating how cancer related concerns and maintaining factors operated to maintain problems.

Discussion

To the best of our knowledge, this study represents the first single-arm feasibility study of individualized face-to-face CBT for AYA survivors of cancer during adolescence and conceptualization of cancer-related concerns reported by the population. The conceptualization is consistent with existing models aimed to guide psychological support for AYA cancer survivors (Haase, 2004; Kazak et al., 2006). Importantly, study findings provide a more comprehensive understanding of the distress experienced by the population than existing models, by integrating data from different sources (e.g., case formulations, clinical notes). As such, study results have important clinical implications for further developing psychological support for the population and expand the growing literature on CBT-based interventions for the population (Campo et al., 2017; Sansom-Daly et al., 2018). However, it should be noted that only 10 participants were recruited raising concerns regarding the feasibility and acceptability of individualized CBT for the population.

Recruitment

Only around 5% of potential participants were included. This finding is in line with research indicating difficulties in recruiting AYA cancer survivors into clinical research (Roth et al., 2016), including trials of psychological interventions (Rabin, Horowitz & Marcus, 2013). Given such recurrent complications, factors that may account for low study inclusion rates should be considered and investigated in future research.

Table 5 Summary of feasibility outcomes.

Outcome	Results	
Recruitment	
Number identified via the Swedish Childhood Registry	n = 286	
Number of potential participants	n = 213	
Number contacted via telephone	n = 111	
Number contacted via post	n = 98	
Number of non-participating		
By exclusion	n = 11	
Declined participation	n = 95	
Lost to recruitment	n = 94	
Number assessed for eligibility	n = 13	
Number found ineligible	n = 3	
Number meeting inclusion criteria	n = 10	
Number included in the study	n = 10	
Reasons for non-participation		
Declined participation	n=95	
No need for support	n = 61	
Online non-consent	n = 10	
Unable to travel to study sites	n = 5	
No interest prior to information given	n = 4	
No interest after receiving information	n = 3	
In counselling	n = 3	
Low trust in psychotherapy	n = 2	
No reason given	n = 2	
No perceived cancer	n = 1	
Intellectual disabilities	n = 1	
Feeling anger	n = 1	
Moved	n = 1	
Lack of time	n = 1	
Lost to recruitment	n=94	
No contact	n = 91	
Administrative failure	n = 2	
No address identified	n = 1	
By exclusion	n=14	
In psychological treatment	n = 6	
Age	n = 2	
In assessment for psychological treatment	n = 3	
Deceased	n = 1	
Unable to travel to study sites	n = 1	
Participation too burdensome	n = 1	
Attrition	10%	
Percentage dropping out of the study	10%	
Percentage dropping out of the treatment		
Data collection	Post-treatment, 70%	
Percentage completing assessments	3-month follow-up, 80%	
Numbers of missing items	n = 6	
Treatment delivery	Mean = 12.6, range = 7–15	
Number of treatment sessions	Mean = 25, range = 18–36	
Length of treatment (weeks)	n = 2	
Number of early treatment terminations		
Number of cancelled treatment sessions	Mean = 3.4, range = 0–11	

Figure 1 Adapted CONSORT diagram illustrating the flow of participants through the study.

Figure 2 Identified precipitatory and current cancer-related concerns and maintaining factors.

Figure 3 Identified precipitatory and current cancer-related concerns and maintaining factors.

Practical barriers to access: A potential explanation for poor recruitment may pertain to practical barriers in accessing support, as the AYA population has a tendency to relocate relatively frequently as well as being occupied with work, school or extracurricular activities. Indeed, cited reasons for non-participation included travel distance and lack of time. Such findings are consistent with barriers identified by previous work examining psychological interventions for AYA cancer survivors, including burden of travel to attend intervention sessions, time limitations, and scheduling difficulties (Kazak et al., 2004; Barnett et al., 2016). Another notable finding was that 98 (47%) of potential participants were provided with study information via post, as opposed to the telephone, due to incorrect or non-identifiable contact details. This concurs with earlier studies on adult survivors of childhood cancer observing recruitment difficulties due to incorrect contact information (Kilsdonk et al., 2015). As many AYAs undergo transitions in housing, employment, and education, which may cause potential changes in preferred contact methods (Barnett et al., 2016), a wider approach including social media and internet-based methods may be more appropriate. This applies to mode of participant communication as well as the initial identification of potential participants, contrary to merely utilizing cancer registries and centers (Gorman et al., 2014). Indeed, social media and internet-based recruitment strategies have been found to be effective in recruitment of cancer (McCusker et al., 2018) and young adult (Musiat et al., 2016) populations. Whilst in-person recruitment (e.g., in-person recruitment at oncology clinics) has been demonstrated as a successful recruitment strategy in comparison to mail-out, telephone-based, and social media recruitment (Rabin, Horowitz & Marcus, 2013), the necessity of approval for site-specific recruitment can cause delays and negatively affect recruitment (Sansom-Daly et al., 2017). In addition, participant’s lack of ability to attend face-to-face therapy may be a reason to explore the potential of internet-administered interventions (Devine et al., 2018; Moody et al., 2015). Offering internet-administered interventions may help overcome some practical barriers to psychological support present in the AYA group.

Helping-seeking behavior: The low participation rate may be explained by findings suggesting that few AYAs show readiness to seek help for psychological distress (Tanielian et al., 2009). Further research suggests that distressed young people struggle to identify, describe, and manage their emotions (Rickwood et al., 2005). Thus, AYAs may not recognize symptoms of psychological distress (Kovandžić et al., 2011). A further barrier to help-seeking is stigma associated with psychological distress and seeking professional help (Clement et al., 2015), as well as viewing the need for psychological support as a sign of weakness (Salaheddin & Mason, 2016). Indeed, difficulties related to embarrassment and low levels of mental health literacy are commonly reported barriers to seeking support among young adults (Gulliver, Griffiths & Christensen, 2010). Given struggles with pity and alienation due to the cancer disease (Lehmann et al., 2014), AYA survivors may be reluctant to identify themselves with the additional stigma related to psychological distress. Internet-administered CBT may represent a solution given its potential for increased anonymity and privacy (Younes et al., 2015) potentially overcoming stigma-related barriers. However, research suggests online solutions are commonly designed to fit available technology as opposed to the needs and preferences of end-users (Hollis et al., 2015). As such, to further increase acceptability and engagement, development of an internet-administered CBT intervention should be informed by representatives of the AYA cancer survivor population to ensure needs and preferences are met.

Cognitive behavioral conceptualization of cancer-related concerns

The novel conceptualization of cancer-related concerns stemming from behavioral case formulations and unstructured interviews extends our knowledge in several areas: cancer-related concerns, and maintaining factors of distress within the population. Present findings point to social avoidance, fear of emotions and bodily symptoms, imbalance in activity, and worry and rumination as significant difficulties to address in psychological interventions for AYA survivors of cancer during adolescence.

Social avoidance

One central implication of clinical significance concerns the social difficulties frequently reported by participants, which may seem unsurprising given that social anxiety is common during late adolescence and young adulthood (Gregory et al., 2007). Further, impaired social functioning (Warner et al., 2016; Wilford et al., 2017) and low levels of social participation (Breuer et al., 2017) have been identified as difficulties among AYA cancer survivors. However, there is to-date little research concerning social participation amongst the population, leaving fear of rejection from peers and avoidance of social situations largely unexplored. Additionally, there is currently no evidence-based psychological intervention for AYA survivors of cancer during adolescence specifically targeting social anxiety. Our findings suggest cancer-related experiences of not feeling liked or accepted by peers, fear of causing discomfort and being a burden, and fear of rejection/exclusion, resulting in the adoption of avoidant coping strategies such as post-mortem rumination, refrain from expressing feelings and needs, and social withdrawal. Such concerns and subsequent behaviors may be vital in understanding the social difficulties experienced by the population. Indeed, targeting social skills deficits among young people experiencing social anxiety have been stressed (Mesa, Le & Beidel, 2015) and appear particularly crucial when working with AYA survivors of brain tumors since the neurocognitive late effects can negatively impact social functioning (Schulte, 2015).

Fear of emotions and bodily symptoms

A second clinical implication for AYA survivors of cancer is based on participants’ reports of health anxiety and refers to a fear of emotions and bodily symptoms, such as intrusive cancer-related thoughts and memories, loneliness and health anxiety, resulting in maintaining factors including avoidance behaviors and obsessions and compulsions. The findings are consistent with literature demonstrating increases in anxiety sensitivity (e.g., fear of anxiety symptoms including body sensations) in adolescents who have experienced stressful life events involving serious illness or death (McLaughlin & Hatzenbuehler, 2009). In addition, fear of cancer recurrence has been found to be higher in AYA cancer survivors than older (Cho & Park, 2017) and mixed-age survivors (Thewes et al., 2017). Further, exposure to health-related stressors, combined with low competence in emotion regulation in fear situations observed in adolescents (Zimmermann & Iwanski, 2014) may increase experiential avoidance (Hayes et al., 1996), which predicts the onset and maintenance of anxiety disorders (Spinhoven, Van Hemert & Penninx, 2017). Accordingly, awareness as well as acknowledgement of fear of emotions and bodily symptoms experienced by AYA cancer survivors may be of importance when providing psychological interventions to the population.

Imbalance in activity

The third clinical implication of study findings relates to AYAs cancer survivors reporting imbalance in activity, which is consistent with the wider literature (Krull et al., 2010). Typical concerns reported were identity difficulties, fear of failure, and lack of routines, with maintaining factors such as lack of engagement with positively reinforcing activities, maladaptive stress behaviors, and procrastination. Activity restriction is associated with impaired mental (Leventhal, 2008) and physical health (Kodama et al., 2009) and may increase the prevalence of late effects in the population (Lipshultz et al., 2015). Whilst low levels of physical activity in AYA cancer survivors are reported, (Bélanger et al., 2011; Hocking et al., 2013) less is known regarding AYA cancer survivors’ disengagement from positively reinforcing activities in other areas. Focusing on increasing opportunities for engagement in positively reinforcing activities, an evidence-based psychological approach like behavioral activation (Ekers et al., 2014) may represent a solution for AYA cancer survivors. Furthermore, behavioral activation can be combined with physical activity promotion (Euteneuer et al., 2017; Pentecost et al., 2015) thus potentially targeting both mental and physical health.

Worry and rumination

In line with a pilot trial of Meta Cognitive Therapy for AYA cancer survivors (Fisher et al., 2015), our findings identified worry and rumination as potentially important targets in the provision of psychological interventions for the population. Key characteristics were worry about the future and fear of cancer recurrence, causing decisional anxiety and depressive as well as existential rumination. Indeed, other research has shown AYA cancer survivors to imagine their future in more illness focused and overgeneralized way, in comparison to AYAs without cancer (Sansom-Daly et al., 2018). Given adaptive problem solving and goal setting are associated with the ability to imagine future events more specifically (Schacter, Addis & Buckner, 2007) such overgeneralized thinking may be an important target for psychological interventions for the AYA cancer survivor population (Sansom-Daly et al., 2018).

Limitations

First, due to most of those approached refraining from study participation, a resultant small sample hampered possibilities of examining potential differences between AYA cancer survivors who were recruited into the study, versus those who declined participation. Still, a preliminary evaluation was conducted and findings highlight important feasibility concerns in the recruitment of AYA cancer survivors. In addition, a number of important feasibility outcomes were not examined, for example, intervention acceptability, acceptability of study outcome measurements, safety, study resources, cost, and therapist training needs and competence (Eldridge et al., 2016; Sansom-Daly et al., 2018). Future studies may benefit from examining a wider range of feasibility outcomes. Further, an a priori set of progression criteria to assess trial feasibility was not established. This is a common limitation in feasibility and pilot trials (Mbuagbaw et al., 2019), resulting in it being difficult to interpret the findings of the present study. When interpreting the cognitive behavioral conceptualization, the impact of study therapists’ and researchers’ background, clinical experience, and theoretical orientation on data collection and analysis should be considered. No formal assessment of therapist competence to the validity of the behavioral case formulations was conducted, and individual case formulations were not empirically evaluated (Mumma & Fluck, 2016). Further, therapist adherence to the CBT model was not assessed. This is of particular importance given fidelity to the intervention may be associated with study outcomes and assist interpretation of findings (Mars et al., 2013). As such, the presented conceptualization of cancer-related concerns and maintaining factors is a hypothesis that could be examined in future studies, whom may benefit from exploring alternative approaches to conceptualizing distress within the population. In addition, member checking (Lincoln & Guba, 1985) to ensure themes generated and subsequent interpretations resonated with interviewees’ experiences was not conducted but could have increased trustworthiness (Krefting, 1991). Further checking to establish trustworthiness with oncology health professionals and parents of AYA cancer survivors could have been conducted. However, it should be noted that a variety of techniques were utilized to establish trustworthiness, including triangulation of both data sources and investigators (Knafl & Breitmayer, 1989). It is unclear whether theoretical saturation and redundancy of data were met (Johnson, Hayes & Wade, 2007). However, given the diversity of human experience, theoretical saturation and redundancy of data may never be achieved and trustworthiness cannot be determined by sample size (Williams & Morrow, 2009). Further research should be undertaken to examine the efficacy of psychological treatment in targeting the cancer-related concerns identified. Finally, although one of the strengths of the current study consists of its efforts to identify cancer-related concerns, it cannot be known whether the same difficulties would have arisen regardless of the cancer experience. Despite these limitations, we believe our findings offer valuable insight for a theoretical understanding of distress among AYA cancer survivors, and the development of psychological support specifically tailored towards this distress.

Future clinical directions

Identification and conceptualization of cancer-related concerns and potential maintaining factors may inform future clinical directions concerning the provision of psychological support to an AYA cancer survivor population. First, evidence increasingly suggests cultural adaptation of CBT is required to work with people from diverse backgrounds (Naeem, 2019). The definition of a cultural group, or subgroup, is not limited to race or religion, but includes factors such as age and physical health status (Naeem, 2019) and thus the AYA cancer survivor population may be considered a cultural group. Given different cultural groups may hold different beliefs concerning the cause and treatment of mental health difficulties and help seeking, it is important to appreciate specific concerns of an AYA cancer survivor population to ensure appropriate CBT adaptation. Indeed, the importance of patient-adapted CBT for the specific needs of an adult cancer population has been identified elsewhere (Poort et al., 2018).

Second, evidence based CBT models for the main difficulties identified in AYA cancer survivors exist. For example for social anxiety disorder (Clark & Wells, 1995; Rapee & Heimberg, 1997), health anxiety (Salkovskis & Warwick, 1986), and worry (Wells, 1999) and as such can inform the development of future interventions for the population. Evidence based CBT techniques for social anxiety may include, but are not limited to, behavioral experiments, attention training, and experiential exercises to understand the negative effects of safety behaviors (Leigh & Clark, 2018) and the addition of social skills training has been associated with improved therapeutic outcomes in children and adolescents (Scaini et al., 2016). Behavioral experiments and graded exposure (e.g., to avoided illness related situations) and response prevention are evidence based CBT techniques that may be utilized for health anxiety (Warwick et al., 1996). Further, behavioral activation is an evidence based technique (Ekers et al., 2014) for depression to facilitate re-engagement in positively reinforcing activities.

Third, findings suggest AYA cancer survivors experience a diverse range of difficulties and concerns, for example social anxiety, fear of cancer reoccurrence, worry and depression. Such findings are in line with high comorbidity rates between anxiety and depressive disorders in adolescent and young adult populations (Essau, 2008) and suggest interventions targeting clusters of symptoms, as opposed to individual diagnoses, (Ranøyen et al., 2018) may be more appropriate. A transdiagnostic conceptualization of distress and intervention approach (Gros, Allan & Szafranski, 2016), targeting underlying maintaining processes (e.g., experiential avoidance and rumination) as opposed to diagnosis specific symptoms (Norton & Paulus, 2016), may be more appropriate (Ander et al., 2018). Indeed, transdiagnostic CBT interventions have shown promise for an adolescent population (Ehrenreich-May et al., 2017) and can be administered via the internet (Newby et al., 2015). Future research may wish to examine the promise of transdiagnostic CBT for an AYA cancer survivor population, including internet-administered interventions.

Conclusions

Whilst able to reveal a number of cancer-related concerns and a conceptualization of these for AYA cancer survivors, our study suffered from a poor recruitment rate, suggesting a need to understand help-seeking behavior in the population. Though there was indication of clinical effectiveness of the intervention used, caution should be taken due to the small sample size. Alternative recruitment strategies for the population appear called for, such as using social media and other internet-based methods. Given practical barriers seen in the population such as lack of time and difficulties of scheduling and attending intervention sessions, internet-administered alternatives may also be more appropriate in terms of support and intervention delivery, in offering larger flexibility than individualized face-to-face CBT. Finally, present results of cancer-related concerns experienced by AYA cancer survivors may prove a step in informing the development of psychological support, tailored to their unique needs.

Supplemental Information

Supplemental Information 1 Dataset detailing the recruitment process

Click here for additional data file.

Supplemental Information 2 Dataset detailing the demographical and clinical outcomes

Click here for additional data file.

We are grateful to Lisa Ljungman for her work as therapists. We would also like to thank Annika Lindahl Norberg for the supervision of qualitative interviews, as well as therapist supervisors Cecilia Arlinger Karlsson and Peter Czatlos.

Additional Information and Declarations

Competing Interests

Author Contributions

Human Ethics

Data Availability

The authors declare there are no competing interests.

Josefin Hagström and Joanne Woodford analyzed the data, prepared figures and/or tables, authored or reviewed drafts of the paper, and approved the final draft.

Malin Ander conceived and designed the experiments, performed the experiments, analyzed the data, prepared figures and/or tables, authored or reviewed drafts of the paper, and approved the final draft.

Martin Cernvall performed the experiments, analyzed the data, authored or reviewed drafts of the paper, and approved the final draft.

Brjánn Ljótsson and Louise von Essen conceived and designed the experiments, performed the experiments, authored or reviewed drafts of the paper, and approved the final draft.

Henrik W. Wiman analyzed the data, authored or reviewed drafts of the paper, and approved the final draft.

The following information was supplied relating to ethical approvals (i.e., approving body and any reference numbers):

Uppsala Regional Ethical Review Board granted Ethical approval to carry out the study within its facilities (Reference number: 2014/443).

The following information was supplied regarding data availability:

The raw inclusion and measurement data are available in the Supplemental Files.

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
