# Peer review of "Heeding the psychological concerns of young cancer survivors: a single-arm feasibility trial of CBT and a cognitive behavioral conceptualization of distress"

_PeerJ, doi:10.7717/peerj.8714_

## Round 0.1 · original submission · Minor Revisions

Thank you for your submission to Peer-J. Both reviewers have provided a number of suggestions which will make the manuscript stronger. Please address these should you wish to resubmit to PeerJ.

·

Basic reporting

This manuscript describes a study in which their primary aims were to examine the feasibility and efficacy of CBT for AYA survivors of cancer and to describe cancer related concerns and maintaining factors for this group. The introduction provides a strong framework for the significance of the study and importance for understanding the treatment needs of the group. The authors did a nice job of explaining their aims and the results and outcomes they examined clearly related to the main purpose. Overall, the writing was clear.

Experimental design

Some strengths of the study design include well-defined aims and questions and the inclusion of a breadth of measure of mental health outcomes and psychological function. However, the article would benefit from providing more detail about their screening procedures and unstructured interviews. Specifically, during the screening assessment (lines 161-170), why did they only screen for depression? Why did they not screen for anxiety or PTSS/PTSD – which are also common in this AYA population? Was that included in the M.I.N.I? Further, more information and detail about the unstructured interviews (described in lines 219-226) would be beneficial. Were there any other standardized questions asked of all participants (besides the one in lines 221-222)?

The authors should also provide support and reasoning for why they did not assess therapist fidelity to CBT. If one of the goals of the paper was to understand efficacy of CBT in this AYA population, it seems that it would be important to show that the treatment participants received was actually CBT.

Most importantly, I think this article would benefit from providing more detail about how the interview data was analyzed (lines 280-288). How did they come up with the themes? Did they use any formal method of analyzing the qualitative interview data (like content analysis) to come up with the four themes? If not, why? Was the interview data only reviewed by members of the research team? Why did they not get input from other sources – oncologists, patients, parents? Asking AYA survivors “what do you think of these themes, do they resonate with you?” would support the validity of the themes. The authors should provide more information as to how they came up with the themes and could provide more support for their approach to this data.

Validity of the findings

The findings of this study provide support for using CBT for AYA cancer survivors, and offer meaningful evidence for providing psychological treatment for this population. However, I am concerned about the strength of the validity of the themes they came up with (see comments above). The manuscript would benefit from the authors providing more of an explanation as to why they think using CBT in this population might be different than the general population. The authors should comment on how CBT can be adapted specifically to incorporate the four cancer-specific concerns and maintaining factors they identified.

The authors did a nice job of presenting their results and conclusion with caution – recognizing that this is a small sample, and may not generalize. They provide support for doing future research in this area.

Additional comments

The article could also be strengthened with more suggestions regarding getting AYA more engaged in psychological treatment, especially considering the low engagement and enrollment. In lines 438-452 the authors explain that AYA don’t often utilize psychological services for a number of reasons. What do the authors then proposed we do to get AYA survivors more willing and engaged in treatment?

Reviewer 2 ·

Basic reporting

1. Overall the paper is well-written and easy to follow. There are one or two spots which would benefit from further editing for clarity, e.g., Results section “previous reception of psychological support” sounds a little awkward

Experimental design

1. The single-arm design of the study is fine given the small numbers, but there is a lack of detail or information provided regarding the sample size that would/should have been expected for this population (from a local context, and also from previous literature where psychological interventions have been offered to survivors). Similarly, there are no hypotheses presented, and although the authors outline that the aim of the study is to establish feasibility, no a priori thresholds for what would constitute reasonable or acceptable levels of feasibility have been presented. This makes it difficult to establish (a) if the findings are consistent with what might have been expected, and/or (b) if what was found justifies the use of this intervention – i.e., is it in fact reasonably feasible? Later in the Discussion the authors state that they were not able to reach data saturation, but again it is not clear what sample they would have needed in order to do so.
2. I also felt that the construct of ‘feasibility’ was defined and examined in fairly narrow terms. There are a number of other aspects of the intervention that could have been examined, either from the AYA participant, or psychologist perspectives, or even ‘system’ perspectives (e.g., cost?). Other elements related to the ‘safety’ of undertaking this type of psychological intervention, with this group, could also have been explored. Clearly these pieces of information are not likely to be available after the fact, but I think some acknowledgement may need to be made in the Limitations section that these aspects were not evaluated. As an example of other ways of establishing/exploring feasibility, please see:
- Sansom-Daly UM, Wakefield CE, McGill BC, Patterson P. (2015). Ethical and clinical challenges in delivering group-based cognitive-behavioural therapy to adolescents and young adults with cancer using videoconferencing technology. Australian Psychologist, Special Issue: Tele-Psychology: Research and Practice, 50(4), 271-278. DOI:10.1111/ap.12112
- Sansom-Daly, U.M., Wakefield, C.E., Bryant, R.A., Patterson, P., Anazodo, A., Butow, P., Sawyer S., McGill, B.C., Evans, H., Cohn, R.J., and The Recapture Life Working Party. (2018). Feasibility, acceptability, and safety of the Recapture Life videoconferencing intervention for adolescent and young adult cancer survivors. Psycho-Oncology, 09 Nov 2018
- Campo RA, Bluth K, Santacroce SJ, et al: A mindful self-compassion videoconference intervention for nationally recruited posttreatment young adult cancer survivors: feasibility, acceptability, and psychosocial outcomes. Support Care Cancer 25:1759-1768, 2017

Validity of the findings

1. In the introduction, the authors highlight two models but then state that these models do not adequately pinpoint mechanisms that may drive/underpin distress. (“…neither model recognizes the full range of cancer-related distress” – what does this mean? In terms of severity? Or types?) I felt that the authors missed an opportunity here to describe in more detail, or unpack, what mechanisms they propose may be driving post-cancer distress, and indeed that they hypothesized they would target via CBT tailored to this group? I realize that some of this information is contained in the CBT conceptualization that is later presented, but in order to have done the study and offered the CBT intervention to these 10 patients I would expect that the authors and therapists would have already had a reasonable idea, at the outset, of what they may be targeting so as to be able to provide a solid therapeutic rationale. There are also other recent papers that have hypothesized about different mechanisms and psychological processes that may contribute to maintaining distress in this population, that may be worth examining to add more nuance to this aspect of your rationale/discussion: See,
- Sansom-Daly, U.M., Wakefield, C.E. (2013). Distress and adjustment among adolescents and young adults with cancer: An empirical and conceptual review. Translational Pediatrics, 2(4):167-197.
- Sansom-Daly, U.M., Wakefield, C.E., Robertson, E.G., McGill, B.C., Wilson, H.L., Bryant, R.A. (2018). Adolescent and young adult cancer survivors' memory and future thinking processes place them at risk for poor mental health. Psycho-Oncology 27(12):2709-2716 - this paper could add to your discussion of identity issues and the way in which not being able to imagine the future impedes goal-setting etc
2. I recognize that this field is constantly evolving, however I think the authors could better situate their intervention in the context of some of the most recent literature, which I believe may be lacking from the present manuscript. For example, recent work has been done by Sansom-Daly and colleagues in Sydney, which also evaluates a CBT focused intervention for AYA cancer survivors. As such, there are a few points in the intro/discussion that may need to be tempered a little to acknowledge that although the present study is novel, it in fact builds on/adds to other work that is also looking at how the CBT model applies to this sort of post-cancer distress, in this group. (e.g., intro: “so far no CBT-based psychological intervention has been developed and tailored for this population” – this is not true).
3. It is a bit unclear how the CBT therapy was ‘sold’ to the participants – how was it presented, what was the rationale presented in terms of what it was targeting/focusing on? As n=61 said they had ‘no need for support’ it would be good to better understand what this actually means. And is it correct that participants did not need to be clinically distressed in order to participate? If this is the case, (and I think this could be made clearer throughout) then it would be helpful for the authors to unpack a little more what it was the survivors thought they were getting/being offered, and for what reason. E.g., were they told it was a coping skills intervention to help them build resilience?
4. Relatedly, if it is true that participants did not need to be clinically-distressed in order to participate, then I’m not completely convinced as to whether the authors’ argument regarding distress and the low rates of opt-ins holds? I think this element of the Discussion may need to be tempered a little bit and with further information regarding what participants were actually told/advertised to inform it.
5. Case formulations are typically tailored to the individual in clinical psychological practice. Consequently, while it is interesting to see the aggregated ‘mechanisms’ that are presented in this paper, it is not clear how clinicians should or could use the aggregate version of all the maintaining factors to inform their practice? It might be informative to add a de-identified worked example of one individual’s case formulation to show how these mechanisms functioned at the individual level to maintain their distress and any other related issues.
6. Relatedly, I think what would add value to this article, particularly for clinician readers, is having a sub-section within your CBT conceptualization section, where after discussion of each ‘mechanism’ you briefly describe what CBT strategies can be used, how they can be tailored, etc to target each mechanism. Presumably this would also involve reflecting on what was used within your own intervention. I think the authors need to be conscious of the fact that within the broader psycho-oncology literature it is not simply agiven or automatically accepted that CBT is the most appropriate therapeutic approach for cancer-related distress (eg., see work by Nick Hulbert-Williams from the UK). I do believe that CBT can play a useful role, and I think the authors have the potential to make a really strong case here with their conceptualization, but I think a little more work is needed to make a strong case as to why the CBT strategies, tailored to the cancer-related mechanisms identified, were highly appropriate and effective. (It would also be useful to know, for the purposes of this paper, to what extent the decision to use various CBT strategies (as in Table 1) was a collaborative decision between AYA-psychologist, versus being solely driven by the therapist. This could speak to the appeal or acceptability of different strategies from AYA perspectives).
7. I also felt that the section where the authors suggest that recruitment may be remedied through the assistance of social media, was not entirely convincing. I think certainly some citations are needed here; while I believe there is some research to suggest this, I think it also depends on the type of study being recruited for, and other reviews have shown that rather in-person recruitment e.g., from a nurse yields the highest success rates in terms of opt-ins. As the authors note, this is an issue that does seem to plague AYA research, particularly intervention studies. The present study seemed to use a single-site design, but there are considerations also to think about regarding where resources are placed in terms of time investment in recruitment – eg., see this recent commentary,
- Sansom-Daly, U.M., Evans, H.E., Ellis, S.J., McGill, B.C., Hetherington, K., Wakefield, C.E. (2017). Something’s got to give: Time-cost trade-offs in site-specific research approval can negatively impact patient recruitment in multi-institutional studies. Internal Medicine Journal, 47(9), 1088-1089.
8. A lot of measures were used, yet there do not seem to be any indices of acceptability or benefit/burden from the perspective of the participants. Can the authors comment on this? Possibly in the limitations?
9. Do the authors have information regarding whether there were any systematic differences between the registry AYAs who opted in vs. out?

Additional comments

I think all my comments have been captured in the above. All the best with the revisions.

---

## Round 0.2 · accepted · Accept

Dear Dr Hagström,

Thank you for your revisions to your manuscript. I have evaluated your revisions and you have addressed the comments - the article is now Acceptable.

With kind regards,
Melanie Zeppel
Academic Editor, PeerJ